# Thermal-healing of lattice defects for high-energy single-crystalline battery cathodes

Shaofeng Li [1,2,7], Guannan Qian[1,3,7], Xiaomei He[4,7], Xiaojing Huang[5], Sang-Jun Lee [1], Zhisen Jiang[1], Yang Yang[5], Wei-Na Wang[3], Dechao Meng[3], Chang Yu[2], Jun-Sik Lee [1], Yong S. Chu[5], Zi-Feng Ma[3], Piero Pianetta [1], Jieshan Qiu [2✉], Linsen Li [3,6✉], Kejie Zhao [4✉] & Yijin Liu [1✉]

Single-crystalline nickel-rich cathodes are a rising candidate with great potential for high-energy lithium-ion batteries due to their superior structural and chemical robustness in comparison with polycrystalline counterparts. Within the single-crystalline cathode materials, the lattice strain and defects have significant impacts on the intercalation chemistry and, therefore, play a key role in determining the macroscopic electrochemical performance. Guided by our predictive theoretical model, we have systematically evaluated the effectiveness of regaining lost capacity by modulating the lattice deformation via an energy-efficient thermal treatment at different chemical states. We demonstrate that the lattice structure recoverability is highly dependent on both the cathode composition and the state of charge, providing clues to relieving the fatigued cathode crystal for sustainable lithium-ion batteries.

[1] Stanford Synchrotron Radiation Lightsource, SLAC National Accelerator Laboratory, Menlo Park, CA 94025, USA. [2] State Key Lab of Fine Chemicals, School of Chemical Engineering, Liaoning Key Lab for Energy Materials and Chemical Engineering, Dalian University of Technology, Dalian 116024, China. [3] Department of Chemical Engineering, School of Chemistry and Chemical Engineering, Frontiers Science Center for Transformative Molecules, Shanghai Jiao Tong University, Shanghai 200240, China. [4] School of Mechanical Engineering, Purdue University, West Lafayette, IN 47906, USA. [5] National Synchrotron Light Source II, Brookhaven National Laboratory, Upton, NY 11973, USA. [6] Shanghai Jiao Tong University Sichuan Research Institute, Chengdu 610213, China. [7] These authors contributed equally: Shaofeng Li, Guannan Qian, Xiaomei He. ✉email: jqiu@dlut.edu.cn; linsenli@sjtu.edu.cn; kjzhao@purdue.edu; liuyijin@slac.stanford.edu

Sustainable and stable high-energy cathode materials are indispensable for the next-generation lithium-ion batteries (LIBs) for a broad range of applications, such as powering long-range electrical vehicles. Ni-rich NMC (LiNi$_x$Mn$_y$Co$_z$O$_2$; $x + y + z \approx 1$, $x \geq 0.6$) with high capacity (>200 mAh g$^{-1}$) has demonstrated great potential as a cathode material for high energy density LIBs[1–3]. Existing commercial NMC cathodes are predominately in the form of micron-sized secondary particles that are agglomerations of nano-sized primary grains[4,5]. There are practical incentives for adopting such a polycrystalline NMC formation. For example, the shortened diffusion length is advantageous to lithium transport and the close packing of the primary grains is beneficial to the energy density. These poly-crystalline NMC materials, however, have abundant grain boundaries and, consequently, suffer from the broadly observed structure degradations, e.g., intergranular and intragranular cracks[6,7], inhomogeneous mechanical strain[8,9], local phase transformation and segregation[10,11]. The anisotropic lattice breathing during the repeated cycling of the energy devices leads to an accumulation of lattice strain and defects, which could be released via particle cracking, to the detriment of the composite cathode electrode's multiscale structural integrity. These cracks create more solid-liquid interfaces that aggravate unwanted side reactions and further exacerbate structural degradation and performance decay. These undesired side reactions are more aggressive in the Ni-rich cathode than in the well-explored NMC compounds with lower Ni contents, e.g., LiNi$_{1/3}$Mn$_{1/3}$Co$_{1/3}$O$_2$[12–15].

A direct and feasible strategy for eliminating the grain-boundary fracturing and for stabilizing the Ni-rich NMC is by using micro-sized single-crystalline particles. This approach eliminates the internal grain boundaries and enables significantly improved cycle performance over the traditional polycrystalline NMC[14,16,17]. Although the intergranular fracturing along grain boundaries is eliminated, the intragranular fracturing caused by the accumulated stresses remains intractable, yet, it still needs to be fine-tuned. It has been reported that reducing the crystal size to below a critical threshold at ~3.5 μm could mitigate the cata-strophic reactions that damage the integrity of the single crystals[18]. However, the localized stresses that are closely corre-lated with the microcrack propagation are pervasive in single-crystalline NMC, which entails further efforts to develop a more practical and effective strategy for addressing the origin of the fatigue damage.

Tuning the battery material properties via controlling the temperature is a viable approach, and has been demonstrated in several different application scenarios. For example, to address the sluggish lithium diffusivity at low temperatures, a battery temperature management system is often implemented for pre-heating the battery before operating it in extreme climates[19]. At a moderately elevated temperature, detrimental effects, such as oxygen release and Li extrusion, have been reported in charged NMC cathodes[20,21]. At a high temperature, the molten salt reactions have been explored for recycling retired battery cathode materials. These battery materials feature a high degree of com-plexity in their responses to the temperature[22,23]. Herein, we formulate a mild thermal treatment method (annealing at 150 °C for a few hours) to address the afore-discussed challenges in defect and strain modulation for single-crystalline NMC cath-odes. Combined with a suite of state-of-art synchrotron techni-ques and theoretical approaches, we demonstrate a thermal-healing of lattice defects in single-crystalline cathodes caused by the thermal-induced release of lattice strain and the structured ordering, which contribute to the capacity restoration.

## Results

**Theoretical prediction and experimental validation.** A set of single-crystalline NMC materials with pre-set stresses as a model system were investigated to reveal how the thermal treatment modulates the structural evolution at the atomic scale and its implications for electrochemical performance. The experimental work is guided by a systematic density functional theory (DFT) modeling that has provided a high-level predictive overview of the thermal effects. The thermal recoverability ($\gamma$) is hereby defined to quantify the effectiveness of recovering the deformed lattice structure through a thermal treatment at 500 K for the NMC series (from NMC333 to NMC811 and at different states of charge) with pre-existing structural defects. More specifically, $\gamma$ is the difference between the distortion indices [Eqs. 1–3 in the Methods and Supplementary Fig. 1] for the prior-to- and post-heating states, respectively. Here, the prior-to-heating state denotes the lattice configuration after annealing at 300 K, while the post-heating state represents the lattice configuration after a thermal cycling proce-dure that involves three steps: (1) heating up the prior-to-heating system to 500 K, (2) annealing at 500 K, and (3) quenching to 300 K. The thermal recoverability is a function of composition and state of charge (SOC), formulating a two-dimensional map (Fig. 1a) with the degree of Li deintercalation, i.e., $x$ in Li$_{1-x}$(NMC)

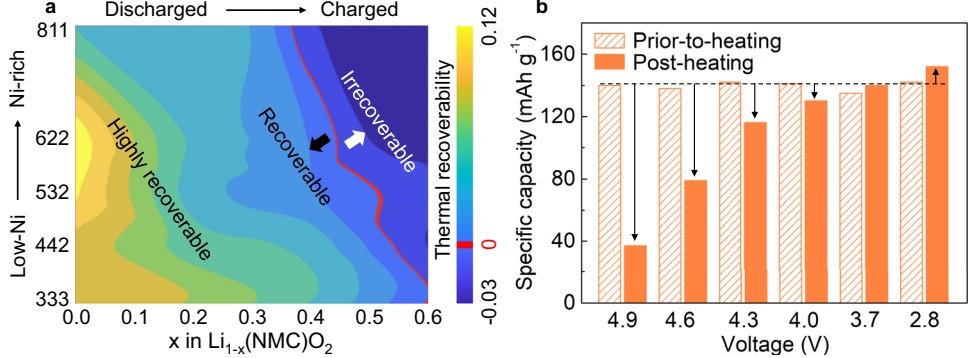

**Fig. 1 Thermal recoverability of single-crystalline cathodes with different Ni and Li⁺ concentrations. a** Diagram showing the effect of Li concentration and NMC compositions on the thermal recoverability. The curved red line in the map annotates the zero-valued boundary. **b** Specific capacity of the NMC622 electrode before and after thermal treatment at different stages. First, all the cells were cycled once with a 0.1 C (1 C = 180 mAh g⁻¹) rate at 2.8–4.9 V vs. Li⁺/Li. Then they were subjected to 10 cycles with a 1 C rate at 2.8–4.9 V to purposely induce lattice defects. After that, the cells were cycled once with a 0.1 C rate at 2.8–4.3 V vs. Li⁺/Li in order to quantify the prior-to-heating capacities. They are then charged to the targeted voltages (2.8, 3.7, 4.0, 4.3, 4.6, and 4.9 V) before being disassembled for the thermal treatment. After the thermal treatment, these electrodes were assembled back into coin cells and were cycled for 2 cycles with a 0.1 C rate at 2.8–4.3 V vs. Li⁺/Li in order to quantify the post-heating capacities.

$O_2$, being the horizontal axis and the Ni concentration being the vertical axis. The curved red line in the map annotates the zero-valued boundary that separates the irrecoverable (with negative $\gamma$ value) and recoverable (with positive $\gamma$ value) zones. More positive $\gamma$ value represents a more effective structural recovery from the corresponding prior-to-heating state that is populated with strain and defects. A negative $\gamma$ value, on the other hand, indicates that a structural deterioration is further induced by the same thermal cycling process. Although the contour lines in Fig. 1a are rather irregularly shaped, two important trends can be observed: (1) $\gamma$ decreases with Li deintercalation, (2) the critical SOC with $\gamma$ equals zero exhibits a clear negative correlation with the Ni concentration. It is useful to note that, in the literature, the thermal stability studies on the NMC cathode family usually highlight the charged state due to their poor structural robustness when a substantial amount of lithium-ion is removed from the lattice matrix[24,25]. In the present work, however, our theoretical modeling of the thermal recoverability suggests the possibility of a thermal-healing effect below certain SOCs.

The DFT prediction of the SOC-dependent thermal healing effect is further corroborated by our experimental validation using single-crystalline NMC622 (Supplementary Fig. 2, Supplementary Table 1), which is one of the most promising candidates among the NMC family. The single-crystalline NMC622 cathodes were first cycled at 2.8–4.9 V to create stresses and then charged to different SOCs via cut-off voltage control (Supplementary Fig. 3, Supplementary Table 2). These cathodes were recovered from the disassembled cells and subjected to thermal treatment before cell re-assembly and electrochemical measurements. The high-voltage abuse (4.9–2.8 V) results in a relatively low capacity of ~140 mAh g$^{-1}$ before our thermal treatment. For a thorough structural characterization and performance assessment of our single-crystalline NMC622 cathodes under normal battery operation conditions, we refer to our previous work[26]. As shown in Fig. 1b, the thermal treatment on the charged electrodes (4.0, 4.3, 4.6, and 4.9 V) causes significant capacity loss, which is in good agreement with conventional wisdom. On the contrary and predicted by our DFT calculations, a noticeable thermal-treatment-induced capacity restoration (~10 mAh g$^{-1}$) is observed in the discharged electrode (2.8 V). It should be pointed out that such a capacity restoration of ~10 mAh g$^{-1}$ clearly exceeds the uncertainty induced by the disassembling-reassembling protocol (merely ~1 mAh g$^{-1}$, Supplementary Table 3) and is reasonable due to the high-voltage abuse executed prior to the cycling and thermal treatment. Some of the detrimental side reactions associated with the electrochemical abuse, e.g., surface phase transition, oxygen release, cation mixing, and active cathode dissolution cannot be reversed through a mild thermal treatment. We purposely designed this mild thermal treatment at 150 °C to isolate the effect of lattice strain and defects repairing from the other complications. Therefore, these results support the concept of thermal recoverability in a single-crystalline cathode.

**Mechanisms of the chemical-state-dependent thermal effects.** To elucidate the underlying mechanism of the chemical-state-dependent thermal healing/damaging effects, we carry out a systematic experimental study as detailed below. Soft X-ray absorption spectroscopy (XAS) was employed to identify the evolution of the valence state in single-crystalline NMC622 electrodes. As shown in the Ni $L_3$-edge XAS spectra (measured in the total electron yield mode with a probing depth of ~5 nm, Supplementary Fig. 4a), the intensity of the low-energy shoulder is increased for the 4.9 V electrode after thermal treatment at 150 °C, indicating the reduction of the surface Ni cations.

However, the 2.8 V electrode shows a different phenomenon, suggesting minor oxidation of the surface Ni cations. Resonant inelastic X-ray scattering (RIXS, with probing depth of ~150 nm) was also conducted to reveal the electronic structure and valence state of the sub-surface Ni cations. As shown in the Ni $L_3$-edge RIXS maps (Fig. 2a), two features centered at excitation energies of 852.5 and 854.5 eV, respectively, exist in both of the initial 2.8 and 4.9 V electrodes. After the thermal treatment, no obvious change was observed for the 2.8 V electrode, suggesting that the valence state of sub-surface Ni cation in the 2.8 V electrode remains mostly unchanged. On the contrary, after the thermal treatment for the 4.9 V electrode, its spectroscopic feature at the excitation energy of 852.5 eV is significantly enhanced in the RIXS map as well as in the corresponding partial fluorescence yields (PFY) spectra (Fig. 2b). This phenomenon originates from the reduction of Ni cations in the subsurface region (up to 200 nm in depth) of the 4.9 V electrode after the thermal treatment, which could be a combined effect of surface reconstruction, phase transformation, and thermally driven outward Li diffusion within the NMC particles[20,22].

To further reveal the bulk electronic and local geometric structures of Ni cations as well as that of Co and Mn cations, X-ray absorption fine structure (XAFS) was performed. As shown in the X-ray absorption near-edge structure (XANES) spectra (Fig. 2c), both Ni and Co K-edge spectra shift to lower energy for the 4.9 V electrode after the thermal treatment, which implies the reduction of the bulk Ni and Co cations. In addition, the oxidation states of Ni, Co, and Mn cations in the 2.8 V electrode remain stable upon the thermal treatment. Further, the Fourier transform (FT) of the extended XAFS (FT-EXAFS) spectra show that the intensities of both M–O, and M–M peaks (M denotes Ni, Co, and Mn) are decreased after the thermal treatment for the 4.9 V electrode (Supplementary Fig. 4b), indicating that more transition metal (TM) cations exhibit an unsaturated coordination environment. The FT-EXAFS spectra also indicate the stable local geometric structure of the 2.8 V electrode. As shown in Fig. 2d and the fitting results (Supplementary Fig. 5, Supplementary Table 4), after the thermal treatment for the 4.9 V electrode, the Ni–O coordination number ($N$ (Ni–O)) and Ni–M coordination number ($N$ (Ni–M)) are decreased from 5.0 to 4.5 and from 5.7 to 4.6, respectively, implying an increase in the oxygen and metal vacancies. Furthermore, the Ni–M atomic distance ($R$ (Ni–M)) is increased from 2.81 to 2.84 Å for the 4.9 V electrode after the thermal treatment, indicative of an increased tensile strain of ~1.1% along the in-plane direction. As for the 2.8 V electrode, $N$ (Ni–O), $N$ (Ni–M), and $R$ (Ni–M) all remain stable under the thermal treatment. As summarized in Supplementary Figs. 6 and 7, Supplementary Tables 5 and 6, the Co and Mn cations also display similar results as the Ni cations for both 2.8 and 4.9 V electrodes. The reduction of Ni and Co cations and the increased vacancies in the 4.9 V electrode after thermal treatment can be attributed to the thermally-induced phase transition that is accompanied by oxygen release and metal migration. Therefore, the dramatically decreased capacity of the 4.9 V electrode after thermal treatment is reasonable and understandable (Fig. 1b). However, the near 10% capacity increment in the 2.8 V electrode after thermal treatment is still inexplicable, which prompted us to further investigate the NMC cathode at the single-particle level, i.e., at the mesoscale[27].

The three-dimensional microstructure and charge distribution of the single-crystalline NMC particles were probed by full-field transmission hard X-ray microscopy (TXM, with a nominal spatial resolution of 30 nm). As illustrated in Fig. 3a, b, no cracks are found in the initial 2.8 and 4.9 V particles, consistent with the improved mechanical robustness of the single-crystalline NMC particles. After the thermal treatment, a distinct crack occurred in

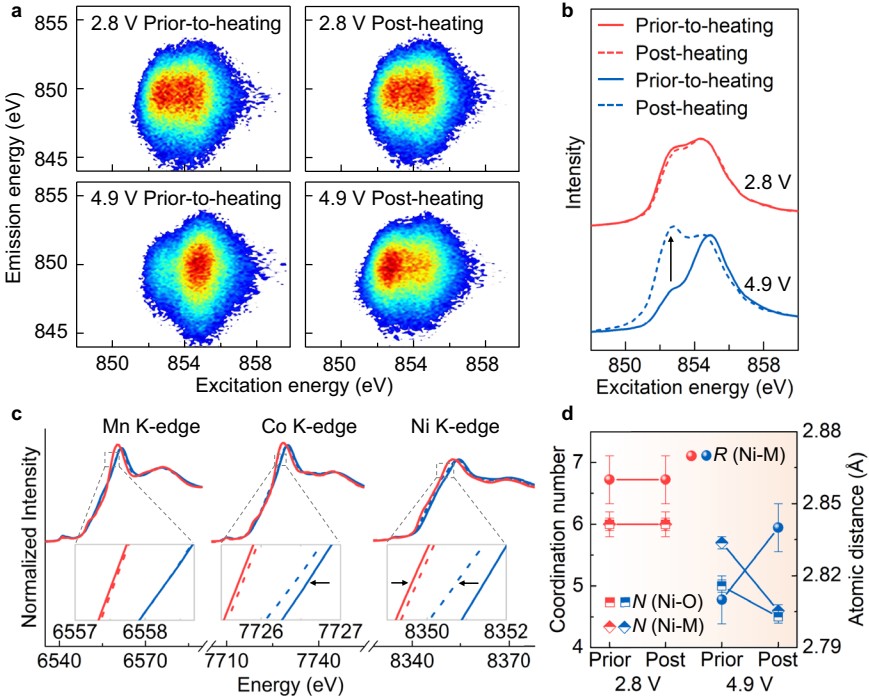

**Fig. 2 Evolution of the electronic and atomic structures in single-crystalline electrodes. a** RIXS maps of the 2.8 and 4.9 V electrodes collected at Ni $L_3$-edge before and after thermal treatment. **b** The PFY spectra extracted from the RIXS maps for the 2.8 and 4.9 V electrodes before and after thermal treatment. **c** The XANES spectra were recorded at Ni, Co, and Mn K-edges for the 2.8 and 4.9 V electrodes before and after thermal treatment. The curve name in (**c**) is identical to (**b**). **d** The coordination number and bond length of the 2.8 and 4.9 V electrodes before and after thermal treatment.

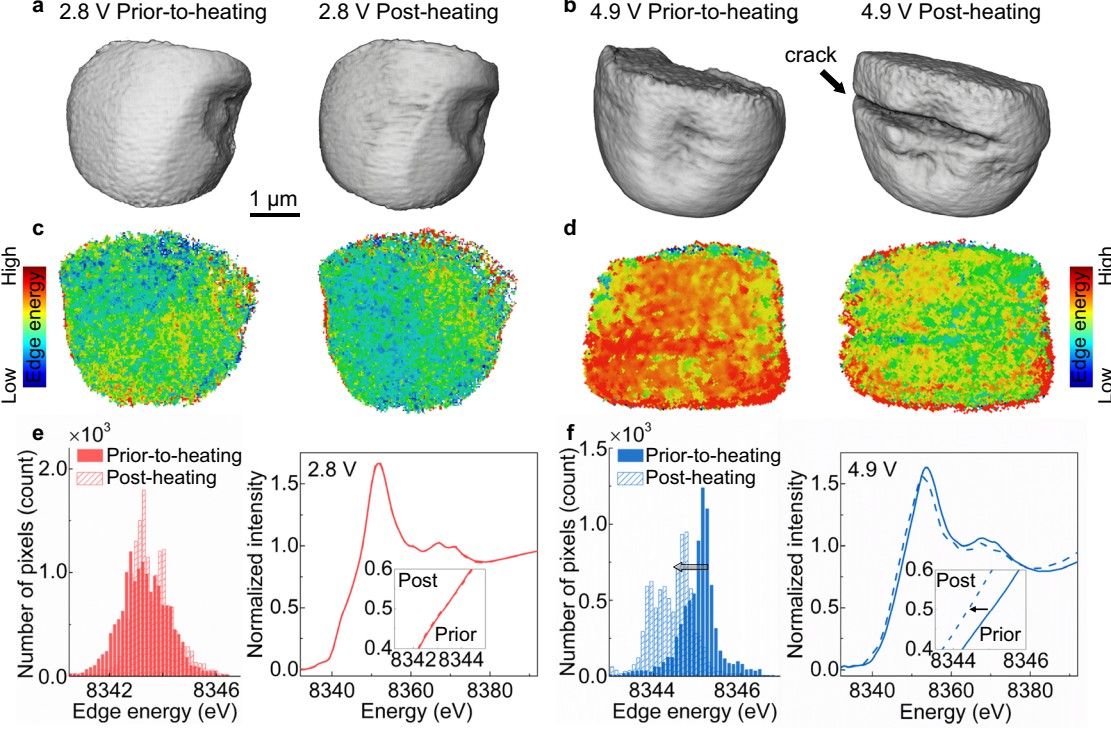

**Fig. 3 Evolution of microstructure and valence state in the single-crystalline NMC622 particle.** The three-dimensional (3D) morphology of 2.8 V (**a**) and 4.9 V (**b**) particles during the thermal treatment. The edge-energy map of 2.8 V (**c**) and 4.9 V (**d**) particles during the thermal treatment. Comparisons of the histogram (left column) and particle-averaged XANES spectra (right column) for 2.8 V (**e**) and 4.9 V (**f**) particles before and after thermal treatment.

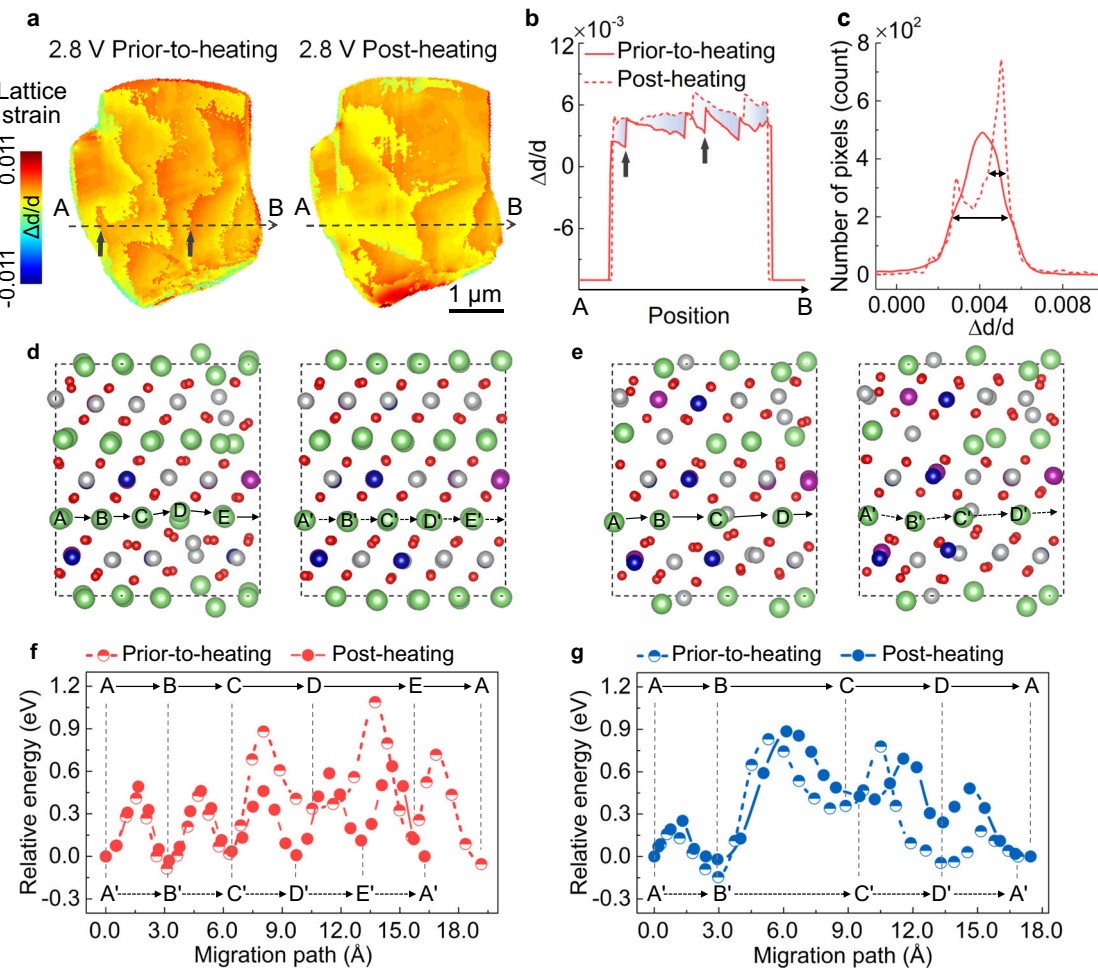

**Fig. 4 The lattice strain evolution of the single-crystalline NMC622 particle. a** The lattice strain map of d-spacing for 2.8 V particles before and after thermal treatment. **b** The line profiles from points A to B as illustrated in (**a**). **c** The relative probability distribution of the lattice strain is displayed in (**a**). The curve name in (**c**) is identical to (**b**). The NMC supercell at 2.8 V (**d**) and 4.9 V (**e**) before and after thermal treatment. The left and right panels in (**d** and **e**) represent before and after thermal treatment, respectively. The green, red, gray, blue, and purple spheres represent Li, O, Ni, Co, and Mn atoms, respectively. Energy profiles at 2.8 V (**f**) and 4.9 V (**g**) along the two Li migration pathways before and after thermal treatment, respectively.

the 4.9 V particle, while the 2.8 V particle maintains its original morphology. The bulk charge distribution can be probed using Ni K-edge energy maps (Fig. 3c, d). A decrement in the Ni oxidation state is clearly observed for the 4.9 V particle after the thermal treatment, e.g., the map color changed from red to green, indicating the lowered Ni K-edge energy value. The histograms and particle-averaged XANES spectra obtained from the spectro-imaging data are shown in Fig. 3e, f. While the Ni K-edge energy of the 2.8 V particle remains steady after the thermal treatment, a Ni reduction is confirmed for the 4.9 V particle, consistent with the insights offered by the Ni K-edge energy maps and the bulk-averaged Ni XANES results. It is also noticeable that the Ni oxidation state distribution in Fig. 3c appears to be more uniform for the 2.8 V particle after thermal treatment, which is also validated by comparing the corresponding histograms that demonstrate a more concentrated Ni K-edge energy distribution after the thermal treatment. It is possible that this results from the Li redistribution during the thermal treatment, which could affect the lattice strain to a certain extent.

**Evolution of lattice strain and Li diffusion kinetics.** For a thorough evaluation of the single-crystalline NMC material at the particle level with high sensitivity to the lattice deformations, we employed scanning XRD with a nano-focal spot of ~30 nm for revealing the lattice strain evolution upon thermal treatment. Figure 4a and Supplementary Fig. 8a, c illustrate three types of lattice distortions (d-spacing inhomogeneity, Y-twisting, and Z-bending) in a 2.8 V NMC particle. The non-uniform maps indicate that the lattice deformation is ubiquitous in the 2.8 V particle even after the first electrochemical cycle. Interestingly, we observe an improvement of the uniformity, especially the uniformity in d-spacing that is associated with the lattice strain, in the 2.8 V particle upon the thermal treatment. Specifically, there are five domains separated by clear boundaries in the d-spacing map of the 2.8 V particle at the prior-to-heating state. The thermal treatment drives these boundaries to migrate and causes the domains to merge, effectively healing the lattice defects. As a result, there are three domains left in the same particle after the thermal treatment at 150 °C. For a better assessment of this observation, the line profiles from points A to B were plotted in Fig. 4b, clearly showing the domain merging phenomenon as indicated by the dis-appearance of two boundaries (see the pointed arrows). The relative probability distributions for all the pixels in the d-spacing maps are presented in Fig. 4c, showing a broader peak in the 2.8 V particle at the prior-to-heating state. Similarly, the relative probability distribution plots for the maps of

Y-twisting and Z-bending also exhibit analogous results (Supplementary Fig. 8b, d).

As illustrated in Supplementary Fig. 9, the 2.8 V particle features a uniform lattice arrangement with suppressed lattice strain after the thermal treatment. This effect could facilitate the transport of Li ions within the single-crystalline NMC lattice with a lower diffusion resistance. For comparison, the lattice strain maps, the corresponding line profiles, and relative probability distribution plots of a 4.9 V particle are shown in Supplementary Figs. 10 and 11. The lattice strain of the 4.9 V particle neither is decreased nor is it homogenized after the thermal treatment. Instead, a thermally induced deterioration can be clearly observed in the Y-twisting maps of the 4.9 V particle. This can possibly be attributed to the local phase transformation from layered to spinel/rock-salt structures. It should be noted that the lattice defects that we are targeting in this study are limited to lattice distortion and plane bending, i.e., d-spacing inhomogeneity, Y-twisting, and Z-bending. It is useful to note that coherence-based imaging methods could offer sensitivities specific to certain types of lattice defects[28,29]. There are pros and cons in both approaches and a more comprehensive comparison of these methodologies is beyond the scope of this paper.

To further elucidate the thermally-induced lattice strain and rearrangement of defects and the impact on the Li diffusion kinetics, we carry out more detailed theoretical calculations. Figure 4d shows the prior-to-heating (left side) and post-heating (right side) supercell structures of NMC622 at the discharged state (2.8 V, fully-lithiated). At the discharged prior-to-heating state, a distinct lattice distortion can be observed, but without any cation mixing due to the fully-occupied Li-layer at the fully discharged state. After the thermal treatment, the pre-existing lattice distortions are largely alleviated and the lattice configuration becomes well ordered. Likewise, the prior-to-heating (left side) and post-heating (right side) supercell structures of NMC622 at the fully charged state (4.9 V, ~60% delithiation) are shown in Fig. 4e. The lattice distortion and cation mixing co-exist in the 4.9 V electrode at the prior-to-heating state. Distinct from the fully discharged scenario, after the same thermal treatment, the structure disorders in the charged prior-to-heating state persist and even intensify. Indeed, the thermally-induced effects for the single-crystalline NMC622 are highly dependent on the SOC.

To further understand this difference and to elucidate its impact on the rate performance, we use the climbing image nudged-elastic-band (CI-NEB) method to examine the Li transport kinetics[30,31]. As shown in Fig. 4f, the average energy barrier ($E_a$) for the prior-to-heating 2.8 V electrode is 0.63 eV (A–B–C–D–E–A paths in Fig. 4d), which is much higher than that of the same electrode after thermal treatment (0.50 eV, A'–B'–C'–D'–E'–A' paths in Fig. 4d). This corresponds to two orders of magnitude difference in the diffusion constant ($D$) at room temperature, e.g., the $D$ value increases from $2.18 \times 10^{-13}$ to $3.32 \times 10^{-11}$ cm$^2$ s$^{-1}$ after thermal treatment. (The diffusion constant is determined by $D = d^2 \nu e^{-E_a/k_B T}$, where $d$ is the hopping distance, $\nu$ is the hopping frequency, $E_a$ is the energy barrier, $k_B$ is the Boltzmann constant, and $T$ is the temperature). Furthermore, the energy barrier varies substantially along A–B–C–D–E–A, for example, the $E_a$ of sub-steps C–D–E is 0.78 eV, which is significantly larger than that of the sub-steps A–B–C and E–A (0.53 eV). These energy barriers are directly linked to the local structures surrounding the migrating Li-ions. As displayed in Fig. 4g, there is no obvious distinction observed in the energy profiles for the 4.9 V electrode before and after thermal treatment, which is consistent with the lattice structure evolution shown in Fig. 4e. The high energy barrier in sub-steps B–C

(B'–C') is caused by the large repulsive force between migrating Li and Ni, which leads to the Li layer migration.

To ensure the statistical representativeness of our particle-level investigation, bulk-averaged synchrotron X-ray diffraction (XRD) measurements on the electrodes were also performed. As shown in Supplementary Fig. 12, the bulk XRD results of the 2.8 and 4.9 V electrodes serve as corroborating evidence for our interpretation based on the nano-diffraction data. To be specific, the intensities of all the diffraction peaks of the 2.8 V electrode are increased upon thermal treatment, indicating an increment of the crystallinity. In contrast, the decreased peak intensity and a shift toward a lower angle was observed for the 4.9 V electrode upon thermal treatment, indicating the formation of substantial lattice defects during the thermal treatment. In connection with the aforementioned analysis of the XAS, TXM, and diffraction images, it can be concluded that the thermally-induced release of lattice strain is the origin of the ~10% capacity increment. The released lattice strain and resulting structural ordering contribute to the recovery of Li transport channels and the capacity restoration. As for the 4.9 V electrode, the increment of TM and oxygen vacancies, particle cracks, and the reduced valence state of TM cations result from the thermal treatment, ultimately, leading to the dramatically decreased capacity.

Herein, we also highlight the significance of our low-temperature healing process for the single-crystalline cathodes in two different perspectives. Firstly, in the real-world battery, the structural hierarchy and chemical complexity dominate the system and could easily overwhelm the other factors. To single out the effect of lattice strain and defects, we design this energy-efficient annealing procedure. Under the mildly elevated temperature, it is reasonable to state that only the lattice strain and defects can be effectively modulated, which is also confirmed by our experimental observations. This approach, therefore, provides a unique opportunity for us to reveal the roles of lattice strain and defects in a real-world battery operation. Secondly, our reported findings are relevant to the battery industry, in particular to the synthesis and recycling of the battery cathode. There are several strategies to reuse the cathode materials in the battery recycling industry, including pyrometallurgy, hydrometallurgy, and direct-recycling[32]. Among them, direct recycling is a promising strategy but is still in its infancy. For this method, the cathode powder needs to be physically separated from the battery first and then processed by adding lithium sources and heating to recover its electrochemical performance. More details can be found in the reported works[33,34], which focus on reducing different damages that appeared in the used cathodes, such as phase transition, microcracks, etc. To reverse the phase transition and recover the layered structure, the normal process is to heat the used cathodes at high temperatures, e.g., 900 °C for NMC622, in oxygen and Li-enriched environment[35]. Our results reported here could be a valuable add-on to the recycling process. After the formation of the particles at high temperatures, the mild-annealing process could be incorporated as the final step to readjust the lattice strain and defects. In our view, these findings serve as valuable seeding efforts and could bring more research attention to this field.

## Discussion

Although the thermal stability of battery cathode is an extensively investigated topic, this work demonstrates two distinct contributions: (i) we report a thermal-healing effect in single-crystalline Ni-rich cathode, which is induced by an energy-efficient annealing process that could potentially be leveraged to improve the lifetime and sustainability of the cathode;

(ii) we systematically evaluate the NMC cathode's thermal recoverability as a function of its composition and SOC, which not only reveals the mechanism but also provides insights for designing the material and the processing strategy.

More specifically, we utilized a DFT modeling approach to predict the thermal recoverability for a series of NMC cathodes with different Ni and Li$^+$ concentrations. Through experimentally quantifying the specific capacity variation upon annealing, a trend of the thermal recoverability for the electrochemically abused NMC622 cathodes was observed unambiguously. For the high SOC samples (charged to 4.9, 4.6, 4.3, and 4.0 V), the significant capacity loss occur after the annealing process at 150 °C, which are derived from the collapse of the layered ($R\bar{3}m$) structure. As the Ni concentration and delithiation degree increase, the layered phase in NMC materials becomes less stable and can be transformed into inactive phases that could impede the lithium diffusion. On the contrary, the samples with low SOC (charged to 3.7 V and discharged to 2.8 V) showed different behavior in response to the same annealing process. In particular, the capacities of the samples at 3.7 and 2.8 V increased after the thermal process, which has largely motivated the thorough investigation reported herein.

In non-Li-rich NMC materials, Ni and Co cations are regarded as the main capacity contributors, whereas Mn cations are very stable and do not undergo redox reactions during the charging/discharging process. As for the oxygen anions, the redox reaction could occur at a high delithiated state, which is associated with the undesired oxygen gas release, cause the host lattice reconstruction, and feature very poor reversibility. In Fig. 2c, the leftshift of the spectra originates from dipole-allowed 1s → 4p electronic transition, indicating the bulk Ni and Co cations in charged NMC622 (4.9 V) are both distinctly reduced upon heating. This is because of the formed NiO-like rock-salt ($Fm\bar{3}m$) structure in the bulk, as echoed by the increase of Ni-M bond distance in Fig. 2d. Meanwhile, as shown in Fig. 3d, f, the thermal-induced Ni reduction of the charged NMC622 (4.9 V) particle is accompanied by an increased redox heterogeneity at the particle level. Moreover, as shown in Supplementary Figs. 10 and 11, the thermal treatment would cause the redistribution of lattice strain within the 4.9 V charged particle. These results suggested that the thermally triggered degradation of the charged NMC622 particles occurs on the surface and in the bulk simultaneously.

For the NMC622 that was discharged to 2.8 V, the valence states of Ni and Co are quite stable (Figs. 2c and 3e) and the coordination environment of Ni cation shows negligible change (Fig. 2d) after the annealing process. While the bulk or beamfootprint averaged spectroscopic signals do not show any noticeable differences after annealing, the rearrangement of the lattice defects is captured in our nano-diffraction experiment at the particle level. The lattice defects can originate from different processes and will play a key role in affecting the cathode performance. For example, Yan et al. reported that dislocations in NMC particles can be incubated by high voltage charging and will lead to strain-induced intragranular cracking[22]. In our cycling protocol, we intentionally choose a high cut-off voltage of 4.9 V to promote the formation of lattice mismatch, deformation, and strain, which will persist even after the cathode is discharged to 2.8 V. After the annealing process, the lattice strain in the discharged particle was partially relieved as evidenced by the disappearance of some strain edges in Fig. 4a.

The reported capacity restoration in layered transition metal oxide cathodes through an energy-efficient thermal annealing process could be functionally important but has been largely overlooked. By systematically probing the thermal recoverability of the NMC cathode as a function of its composition and SOC using, our work not only reveals the mechanism behind the thermal healing of the layered cathode but also offers new sights on developing self-healing battery materials to promote the sustainability of LIBs.

## Methods

**Material synthesis.** The LiNi$_{0.6}$Co$_{0.2}$Mn$_{0.2}$O$_2$ was synthesized via a molten-salt assisted method reported in our previous study[26,35]. LiOH, Li$_2$SO$_4$, and ethanol were purchased from Adamas. The Ni$_{0.6}$Co$_{0.2}$Mn$_{0.2}$(OH)$_2$ was acquired from the GEM Co., Ltd. The Ni$_{0.6}$Co$_{0.2}$Mn$_{0.2}$(OH)$_2$ was mixed with LiOH, Li$_2$SO$_4$ with a molar ratio of 2:3:1 by grinding. The mixture was transferred into a cylindrical alundum crucible covered by a lid and heated to 950 °C with a ramping rate of 15 °C min$^{-1}$ for 3 h and then turn down to 900 °C for 10 h with a rate of 2 °C min$^{-1}$ in oxygen atmosphere before cooling down to 100 °C with a rate of 3 °C min$^{-1}$. The reaction mixture was removed from the excess Li-salts by washing with deionized water. The collected powder was dried at 80 °C in the air for 2 h before being thermally treated at 750 °C for 6 h in an oxygen atmosphere. Finally, the powder was passed through a 400-mesh sieve before testing.

**Electrochemical test.** The LiNi$_{0.6}$Co$_{0.2}$Mn$_{0.2}$O$_2$, carbon black (Shanghai SJ-htech. Inc.), and polyvinylidene fluoride (PVDF, Solvay 5310) were mixed with a weight ratio of 90:5:5 in N-methyl-2-pyrrolidinone (NMP, Adamas) by a Thinky Mixer (ARE-310). The slurry was coated onto the aluminum foil before drying at 70 °C for 2 h. The electrode was further dried at 120 °C in a vacuum for 12 h. The active loading is about 4–5 mg cm$^{-2}$. The 2032-type coin cells were assembled using LiNi$_{0.6}$Co$_{0.2}$Mn$_{0.2}$O$_2$ cathode, Li-metal anode (Shanghai SJ-htech. Inc.), polyethylene (PE, SENIOR-SW16) separator, and electrolyte (1.1 M LiPF$_6$ dissolved in ethylene carbonate and ethyl methyl carbonate with a weight ratio of 3:7, Shanghai SJ-htech. Inc.) in Ar-filled glovebox.

The galvanostatic cycling was performed using a battery cycler (Shenzhen Neware, BTS4000-5 V, 10/1.0 mA version). For electrochemical thermal recovery experiments, the active loading is about 1.5–2 mg cm$^{-2}$. The cells were cycled 1 cycle with 0.1 C (1 C = 180 mAh g$^{-1}$) rate at 2.8–4.9 V vs. Li$^+$/Li before cycling 10 cycles with 1 C rate at 2.8–4.9 V. Then the cells were cycled 1 cycle with 0.1 C rate at 2.8–4.3 V vs. Li$^+$/Li before charging to specific voltages (2.8, 3.7, 4.0, 4.3, and 4.6 V). The NMC electrodes were separated from the cells, washed by dimethyl carbonate (DMC) to remove the residue electrolyte, and dried in Ar-filled glovebox. The NMC electrodes were heated at 150 °C for 2 h in the air before assembling the new cells. The new cells were cycled 2 cycles with a 0.1 C rate at 2.8–4.3 V vs. Li$^+$/Li. For the XRD, Soft XAS, XAFS, TXM, and scanning X-ray probe measurements, the active loading of the electrode is about 5–6 mg cm$^{-2}$. The cells were charged to 4.9 V or cycled 1 cycle 2.8–4.9 V with a 0.1 C rate. The NMC electrodes were separated from the cells, washed by DMC to remove the residue electrolyte, and dried in Ar-filled glovebox. For this study, there are two competing factors that drive our selection of the thermal treatment temperatures: (1) higher temperature can serve as a more effective perturbation of the lattice rearrangement; (2) lower temperature is more desirable from the energy efficiency and side-reaction-suppression perspectives. In our experiment, after a few trial-and-error attempts, we settled to 150 °C. We declare that further efforts are needed to truly pin down the optimal temperature. The optimization of the process is an ongoing effort, but the fundamental mechanisms are the same as those reported in this paper.

**Characterization.** Scanning electron microscopy (SEM) was performed using a Phenom Pro microscope. Inductively coupled plasma-atomic emission spectrometry (ICP-AES) was carried out on an iCAP™ 7600 ICP-OES Analyzer (Thermo Fisher). Focused ion beam (FIB)/SEM imaging and scanning transmission electron microscopy (STEM) specimen preparation were conducted on a Zeiss Crossbeam 540. The FIB-prepared samples were investigated by a Cs-corrected JEOL JEM-ARM200F operated at 200 kV. Synchrotron XRD was conducted at beamline 13-1 of Stanford Synchrotron Radiation Lightsource (SSRL) with the beam energy of 12.7 keV and wavelength of 0.975 Å.

**Soft XAS and XAFS Measurements.** Soft XAS measurements were conducted at beamline 10-1 of SSRL. The incident beam was monochromatized by a 600 lines mm$^{-1}$ spherical grating monochromator, and the incident angle was set as 30° from the sample surface. All the XAS spectra were normalized by the intensity of the incoming beam, which was measured as a drain current on an electrically isolated gold-coated mesh simultaneously. RIXS measurements were performed using a transition edge sensor (TES) spectrometer, which consisted of a 240-channel energy-dispersive detector array[36,37]. The energy collected by the TES was calibrated through separate measurements of a reference sample consisting of C, N, O, and various 3d transition metal oxides with known emission energies.

Hard XAS measurements, including XANES and EXAFS, were performed at beamline 4-1 of SSRL. The Ni, Co, and Mn foils were used to calibrate all the XAFS spectra. The XANES spectra were processed using the ATHENA software package, and the ARTEMIS module of IFEFFIT was employed to do EXAFS fitting analysis[38].

**TXM measurements**. The TXM experiments were carried out at beamline 6-2c of SSRL[39]. The single-crystalline particles were shaved off from the electrode and loaded into a quartz capillary in the Ar-filled glovebox. The quartz capillary was then mounted on the sample holder and kept perpendicular to the incident X-ray beam. During the experiment, the sample was placed under a slow and steady helium flow. The nano-tomography data were collected by rotating the sample holder from $-90°$ to $90°$ with an angle step size of $0.5°$ at an incoming X-ray energy of 8800 eV. The Ni K-edge 2D map was recorded by taking projection images with an energy scan from 8100 to 8800 eV in 134 steps (0.5 s exposure time, 10 repetitions, binning 2, $1024 \times 1024$ pixels). The pixel size of TXM images varies as a function of X-ray energy, and all the images are scaled to match the data at 8800 eV with a pixel size of 34.3 nm. The data analysis was performed using in-house-developed software known as TXM-Wizard[40].

**Scanning X-ray probe measurements**. The scanning X-ray probe measurements were implemented at the hard X-ray nanoprobe beamline 3-ID of National Synchrotron Light Source II with the beam energy of 9 keV and wavelength of 1.378 Å[41–44]. The single crystal was rotated over a 180 range with a XRD detector recording the diffraction pattern at each rotation angle in order to locate the target Bragg peak. A pixel array detector was then oriented to measure the strongest (104) peak. The crystal was rocked over a $2°$ angular range in the vicinity of the (104) Bragg peak, and a two-dimensional raster scan was conducted at each rocking angle. The local Bragg diffraction measurements were performed in sync with the raster scan, and the raster scans were repeated for a series of rocking angles with diffraction signals above the noise level.

**Modeling methods**. The perfect NMC supercell is constructed with $R\bar{3}m$ space group, where Li, TM, and O occupy the 3b, 3a, and 6c sites. Each fully lithiated NMC composition contains 120 atoms. We adopted the perturbation method to generate the structural defects. The original regular octahedra of $LiO_6$ and $TMO_6$ are deformed to induce the disordered structure, including Li (TM)-polyhedral, voids, cation mixing, and altered coordination numbers. The degree of perturbation plays a vital role. If the perturbative amplitude is too large (for example, in the case of a phase transition), the structural defects cannot be healed by the mild thermal treatment. In this study, we construct the perturbation model as follows: Li layers and TM layers are tilted $11.5°$ counterclockwise relative to the $b$-axis; the O layers are perturbed along the $c$-axis following a sine wave function with an amplitude of $A \approx 0.7$ Å. At the fully lithiated state, initial perturbation models are the same for each NMC composition. During delithiation, Li atoms are randomly and consecutively removed from the lattice structure with a stabilized perturbation model. DFT calculations and DFT-based AIMD simulations are performed using the Vienna Ab initio Simulation Package[45]. The projector augmented wave method[46] is implemented with an energy cutoff of 520 eV to describe the ion-electron interaction. The exchange-correlation energy functional Perdew–Burke–Ernzerh (PBE) with generalized gradient approximation is employed[47]. Before defects are introduced, the Brillouin zone is sampled by $3 \times 3 \times 1$ $k$-points in the Monkhorst–Pack scheme, lattice constants, and atomic coordinates are fully optimized with an energy convergence criterion of $10^{-5}$ eV per atom and a force convergence criterion of 0.04 eV Å$^{-1}$. The DFT $+ U$ method was used to correct the Coulombic repulsion of the TM-3$d$ states, combining van der Waals (vdW) corrections (DFT $+$ D3) with DFT $+ U$ to yield highly consistent lattice parameters compared to the experimental results[48]. The Hubbard $U$ values for Ni, Mn, and Co are taken from previous studies[49,50], with $U–J$ being 6.70, 4.20, and 4.91 eV, respectively. In AIMD modeling, the NVT ensemble with a Nose–Hoover thermostat is used to regulate the temperature. The time-step $\Delta t = 2$ fs and single $\Gamma$ point Brillouin zone sampling are used. AIMD is first conducted at $T = 300$ K for 6 ps to relax the perturbation model until it is totally stabilized, after this step, we obtain the prior-to-heating structure. Next, the stabilized structures are sequentially heated up to 500 K with the rate of $3.33 \times 10^{13}$ K s$^{-1}$ followed by 4 ps of annealing at $T = 500$ K to equilibrate the system. Finally, the systems are quenched back to 300 K with a cooling rate of $1 \times 10^{14}$ K s$^{-1}$, and the post-heating structure is prepared.

Li diffusion and the corresponding energy barriers are calculated using the CI-NEB calculations, complementary to the NEB method[30,31]. CI-NEB calculations are carried out with the standard PBE functional (without $+U$) to avoid the over-localization of electron density between the diffusion barrier and the charge transfer barrier[51–54]. Dispersion corrections are considered for the CI-NEB calculations since van der Waals (vdW) corrections are important at low Li concentrations[51]. The lattice parameters for CI-NEB calculations are fixed. In addition, to avoid local structures collapsing and drifting, atoms more than 5 Å away from migrating Li in all configurations are fixed.

After the thermal remediation process, some of the defects disappeared or partially recovered, while some of them remain or become more disordered.

To assess the variation of the structural defects in the whole structures for prior-to- and post-heating, the average of all local distortion represents the overall structural defect level. Therefore, we introduce the local distortion index ($d$) of Li-polyhedra and TM-polyhedra using the following equation:

$$d = \frac{1}{n}\sum_{i=1}^{n}\left(\frac{|\theta_i - \theta_{ave}|}{\theta_{ave}}\right) \qquad (1)$$

where $\theta_i$ is the O–Li–O (O–TM–O) dihedral angle for the central Li with the $i$th pair of the two neighboring O atoms, $\theta_{ave} = 90°$ is the average O–Li–O (O–TM–O) angle in the regular $LiO_6$ ($TMO_6$) octahedra. For $d$ calculation, the cut-off distance for O atoms surrounding Li (TM) atoms is less than 2.8 Å.

Each prior-to-heating system is used as a reference to compare to the corresponding post-heating state. Due to the temperature effect, smaller thermal fluctuations triggered distortion should be filtered. Here, $d = 0.08$ as a threshold, lower $d$ values will be neglected, so the average distortion index $d_{ave}$ calculated by the following equation:

$$d_{ave} = \frac{1}{k}\sum_{i=1}^{k}d_i \qquad (2)$$

where $k$ is the number of $d \geq 0.08$, and $d_i$ is the $i$th value of $d \geq 0.08$ for the prior-to-heating state of each composition at a given delithiation degree. The smaller $d_{ave}$ represents the higher-ordered lattice structure.

Thermal recoverability ($\gamma$) is the distortion index difference between the post- and prior-to-heating structures, which is calculated by the following equation:

$$\gamma = -\left(d_{ave-post-heating} - d_{ave-prior-to-heating}\right) \qquad (3)$$

where higher positive $\gamma$ value means a higher recoverability, while a more negative $\gamma$ value means the structural deterioration after thermal treatment.

## Data availability

The data that support the plots within this paper and another finding of this study are available from the corresponding author upon reasonable request.

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

## Acknowledgements

Use of SSRL at SLAC National Accelerator Laboratory is supported by the U.S. Department of Energy (DOE), Office of Science, Office of Basic Energy Sciences under contract number DE-AC02-76SF00515. The use of the Hard X-ray nanoprobe (HXN) beamline at the National Synchrotron Light Source II (NSLS-II) is supported by the U.S. DOE. NSLS-II is an Office of Science user facility operated by Brookhaven National Laboratory under contract number DE-SC0012704. K.Z. acknowledges the support of the National Science Foundation through the grants CMMI-1726392 and DMR-1832707. This work was partly supported by the National Natural Science Foundation of China (Nos. 22008154, U2003216) and the Sichuan Science and Technology Program (2021JDRC0015). This work is also supported by excellent Ph.D. graduates' development grants of Shanghai Jiao Tong University (to G.Q.). The authors acknowledge the experiment support from R. Davis for the XAFS experiment at beamline 4-1 of SSRL, and from Ajith Pattammanttel for the nano-diffraction experiment support at HXN 3-ID beamline in NSLS-II. The engineering support from D. Van Campen, D. Day, and V. Borzenets for the TXM experiment at beamline 6-2c of SSRL.

## Author contributions

Y.L., K.Z., L.L. and J.Q. conceived the study. S.L. collected the XAS data with the help of S.-J.L. and J.-S.L. S.L., X.H., Z.J. and Y.Y. performed the nano-probe experiments. S.L. and Z.J. conducted the in situ TXM experiments. G.Q., W.-N.W. and D.M. synthesized single-crystal samples and conducted the electrochemical measurements. X.H. and K.Z. carried out the theoretical calculation. C.Y., Y.S.C., Z.-F.M. and P.P. contributed to the data analysis and discussion. S.L. and Y.L. drafted the paper with input from all coauthors. S.L., G.Q. and X.H. contributed equally to this work.

## Competing interests

The authors declare no competing interests.
