## [Peer Review File · Nature Communications]

REVIEWER COMMENTS

Reviewer #1 (Remarks to the Author):

The manuscript entitled „Thermal-healing of lattice defects for high-energy single-crystalline battery cathodes” reports a path to recover lattice structure of a Ni-rich NMC battery material by decreasing lattice strain through a simple heat treatment. The authors employed DFT modeling to predict thermal recoverability, then experimentally demonstrated. Various synchrotron data were collected on cathodes with two very different SOCs to detect how oxidation state and lattice strain varied before and after heat treatment.

All the results shown are noteworthy, carefully analyzed and interpreted. The work is significant in the field to understand how Ni-rich NMC cathode materials could be recovered and how their defective structure vary with SOC and heat treatment. The work extensively support conclusions and claims. The work meets and exceed the expected standard in the field. The manuscript requires a few minor revisions.

Further experimental details are needed for the work to be reproduced: the supplier and purity of reactants and of all the cell components should be reported (e. g., which supplier and product among PVDF was used to prepare the electrodes, which PE was used as separator and so on).

In line 68 of page 4 the authors use the expression “Li extrusion”. Does it mean Li extraction or deintercalation? Please clarify.

The “Modeling methods” section contains a lot of typos (absolutely absent in the rest of the manuscript). Please use the spelling corrector of any word processor to fix them. The English language used is, otherwise, appropriate and correct.

Reviewer #2 (Remarks to the Author):

In this manuscript, the authors investigate how single-crystal NMC cathode materials respond to thermal treatment using a variety of X-ray techniques. The manuscript is well-organized and reports interesting new results. I suggest that it may be considered for publication after some minor changes.

(1) In page 5, line 85, the authors mentioned “deformed lattice structure”. More explanation is needed to clarify what it is. Does it involve structural rearrangements such as the well-known layered/spinel/rocksalt phase transitions? Supercells (4.9 V sample) are shown in Figure 4e. What do the different colored spheres represent? How are they constructed?

(2) The authors need to explain how they chose the thermal treatment temperature? Is it possible to use a temperature lower than 500 K?

(3) More explanation and the detailed measurement conditions are needed in the caption of Figure 1b. It is very difficult to understand the history of the electrodes shown in Figure 1b.

(4) In page 9, line 176, I am wondering why the investigation at the single particle level equals to “at the mesoscale”. This does not seem straightforward to me.

(5) In the 4.9 V sample (prior-to-heating), did the authors observe layer to spinel/rocksalt transitions?

(6) In Figure 4a to 4c, the average $\Delta d/d$ actually increases after heating. Why would this happen?

(7) It is recommended that the author specify the wavelength of synchrotron X-ray diffraction measurements.

Reviewer #3 (Remarks to the Author):

In this manuscript, the authors studied the thermal effect on the single-crystal NMC cathode particles. The thermal healing behavior at low SOC seems quite interesting, which suggests a new strategy to extend battery cycle life. This work was done by a group of scientists with a lot of experience on battery materials and characterization techniques. The manuscript is well-written. It may be published after the authors address the following minor issues.

1. The authors need to clarify what kinds lattice defects they are trying to “heal”? Do they just mean lattice distortion and plane bending? or line/screw dislocations are also included.

2. Is there a specific reason to choose 150 degree C for the thermal treatment?

3. In Figure 3b, the authors did not observe cracks in the 4.9 V prior to heating particle. This is somewhat surprising to me. In several previous papers, such as Pengfei Yan et al, Nature Commun 2017, 8, 14101. Intragranular fracture was observed in NMC333 particles charged to 4.7 V. Can the authors explain what led to such difference?

4. The authors mentioned “the localized stresses and dislocations that are closely correlated with the microcrack propagation are pervasive in single-crystalline NMC” ..Did they observe dislocations and their evolution like those shown in Andrew Ulvestad et al, Science, 2015, 348, 6241?

Reviewer #4 (Remarks to the Author):

The authors report on a mild-annealing process at temperatures around 500 K for healing lattice strains and defects in single-crystalline NCM cathode particles. It is claimed that this method works only in specific state-of-charge regimes at voltages around 2.8 V. XAS and XANES measurements indicate that the thermal annealing at 2.8 V. does not lead to changes in the valence state of surface Ni cations, whereas at voltages around 4.9 V, annealing leads to a reduction of Ni cations in subsurface regions, which lowers the particle capacity. By means of DFT and CI-NEB calculations, activation barriers for Li transport are calculated, and it is shown that lattice healing during mild annealing enhances the Li diffusion coefficients by about two orders of magnitude. Finally, the authors suggest that during battery recycling, the mild-annealing process might be carried out as a final step after the normal high-temperature heating of NMC.

While the work is potentially very interesting for the battery community, I have serious problem in understanding the cycling protocols before and after mild annealing as well as the reported results for the capacities:

- Before annealing, all cathodes were charged up to 4.9 V for several cycles. This is an extremely high potential, which should lead to massive electrolyte decomposition. In commercial batteries, NMC cathodes are never charged to such high potentials. Please comment on this.
- After annealing, the cathodes were charged only up to 4.3 V, and Coulomb efficiencies significantly larger than 100% were observed. The extreme case is Fig. S3e with a charge capacity close to zero for 1st post-to-heating, but a discharge capacity of 90 mAh/g. Furthermore it is strange that all prior-to-heating curves from 3.7 to 4.9 show a discharge capacity around 140 mAh/g, while the curve at 2.8 V shows only 125 mAh/g. Overall, it seems problematic to deconvolute the influence of mild annealing from the influence of the distinct cut-off potentials. Please comment on these issues.
- The discharge capacities in Fig. 3 a-f differ from those in Tab. S2, in particular for the cathode “heated at 2.8 V”. Why is this?

Furthermore, it is unclear why the material loading was only 1.5-2 mg. This corresponds to a cathode thickness in the range of only 5 μm , much lower than the typical cathode thicknesses of 80-100 μm in typical lab-scale and commercial batteries. Please comment on this.

Minor points:

- What is the origin of Ni cation reduction at high voltages of 4.9 V? Release of molecular oxygen?
- Page 9: The authors use the term “operando TXM”. However in this case, the method was not used in operando.

In summary, major revisions of the manuscript are needed, before a decision about the acceptance can be made.

Point-to-point revision summary for

Article NCOMMS-21-35847–*Thermal-healing of lattice defects for high-energy single-crystalline battery cathodes*

Reviewer comments are color-coded in black

Author responses are color-coded in blue

Manuscript revisions are color-coded in red

We are thankful to all the four reviewers for their efforts in evaluating our submission and for making constructive comments, which we find very useful for our revision. We are delighted that the first three reviewers have recommended the acceptance of our manuscript for publication in Nature Communications. Reviewer #4 raised specific questions on the electrochemical protocols, which is important information to include. All the specific suggestions are now addressed in details. We believe that this review process has made our manuscript stronger and we are very grateful for this. Our point-to-point response is summarized below and we look forward to further interactions with all the reviewers and the editorial office.

Reviewer #1:

The manuscript entitled “Thermal-healing of lattice defects for high-energy single-crystalline battery cathodes” reports a path to recover lattice structure of a Ni-rich NMC battery material by decreasing lattice strain through a simple heat treatment. The authors employed DFT modeling to predict thermal recoverability, then experimentally demonstrated. Various synchrotron data were collected on cathodes with two very different SOC to detect how oxidation state and lattice strain varied before and after heat treatment.

All the results shown are noteworthy, carefully analyzed and interpreted. The work is significant in the field to understand how Ni-rich NMC cathode materials could be recovered and how their defective structure vary with SOC and heat treatment. The work extensively support conclusions and claims. The work meets and exceed the

expected standard in the field. The manuscript requires a few minor revisions.

Response:

We are thankful for the positive and encouraging assessment of our work. We also appreciate the specific comments listed below.

1. Further experimental details are needed for the work to be reproduced: the supplier and purity of reactants and of all the cell components should be reported (e.g., which supplier and product among PVDF was used to prepare the electrodes, which PE was used as separator and so on).

Response:

In our revision, we have added the experimental details regarding the supplier and purity of reactants and all the cell components.

Revision:

In the “Method” section, we have added the following details in the “Electrochemical Test” part.

“The $\text{LiNi}_{0.6}\text{Co}_{0.2}\text{Mn}_{0.2}\text{O}_2$, carbon black (Shanghai SJ-htech. Inc.), and polyvinylidene fluoride (PVDF, Solvay 5310) were mixed with a weight ratio of 90:5:5 in N-methyl-2-pyrrolidinone (NMP, Adamas) by a Thinky Mixer (ARE-310). The slurry was coated onto the aluminum foil before drying at 70 °C for 2 h. The electrode was further dried at 120 °C in vacuum for 12 h. The active loading is about 4-5 mg cm⁻². The 2032-type coin cells were assembled using $\text{LiNi}_{0.6}\text{Co}_{0.2}\text{Mn}_{0.2}\text{O}_2$ cathode, Li-metal anode (Shanghai SJ-htech. Inc.), polyethylene (PE, SENIOR-SW16) separator, and electrolyte (1.1 M LiPF_6 dissolved in ethylene carbonate (EC) and ethyl methyl carbonate (EMC) with a weight ratio of 3:7, Shanghai SJ-htech. Inc.) in Ar-filled glovebox.”

2. In line 68 of page 4 the authors use the expression “Li extrusion”. Does it mean Li extraction or deintercalation? Please clarify.

Response:

By “Li extrusion” we are referring to the effect of heat-induced extraction of Li from the cathode material, forming Li-based whisker structure. This phenomenon was reported in our earlier publication [shown in **Figure R1**, *J. Mater. Chem. A* 6, 23055-23061 (2018)]. To avoid misunderstanding, we have revised it to “Li extraction” in our revision.

Figure R1. Thermally driven (under 380 °C) Li-based whisker growth on the surface of hollow spherical $\text{LiNi}_{0.4}\text{Mn}_{0.4}\text{Co}_{0.2}\text{O}_2$ particles. Figure from *J. Mater. Chem. A* 6, 23055-23061 (2018).

3. The “Modeling Methods” section contains a lot of typos (absolutely absent in the rest of the manuscript). Please use the spelling corrector of any word processor to fix them. The English language used is, otherwise, appropriate and correct.

Response:

We thank the reviewer for pointing this out. In our revision, we have corrected all the typos in the “Modeling Methods” section.

Reviewer #2:

In this manuscript, the authors investigate how single-crystal NMC cathode materials respond to thermal treatment using a variety of X-ray techniques. The manuscript is well-organized and reports interesting new results. I suggest that it may be considered for publication after some minor changes.

Response:

We thank the reviewer #2 for the positive assessment and for recommending the publication of our manuscript in *Nature Communications*.

1. (1) In page 5, line 85, the authors mentioned “deformed lattice structure”. More explanation is needed to clarify what it is. Does it involve structural rearrangements such as the well-known layered/spinel/rock-salt phase transitions?

Response:

The “deformed lattice structure” here means “lattice structure with pre-existing structural defects”. It could include inhomogeneity in d-spacing, twisting and bending of the lattice matrix, etc. The severe structural rearrangement, i.e., the layered/spinel/rock-salt phase transitions, are not the focus in this context because they cannot be healed by the mild thermal treatment. A much higher temperature is needed to facilitate the recrystallization of the material, which is often used in the synthesis/recycle process and is not the focus of this paper.

Revision:

According to the reviewer’s suggestion, we have clarified this in the “Modeling Methods” part of the “Methods” section.

“We adopted the perturbation method to generate the structural defects. The original regular octahedra of LiO_6 and TMO_6 are deformed to induce the disordered structure, including Li (TM)-polyhedral, voids, cation mixing, and altered coordination numbers. The degree of perturbation plays a vital role. If the perturbative amplitude is too large (for example, in the case a phase transition), the structural defects cannot be healed by the mild thermal treatment. In this study, we construct the perturbation

model as following: Li layers and TM layers are tilted 11.5° counterclockwise relative to the b-axis; the O layers are perturbed along c-axis following a sine wave function with an amplitude of $A \approx 0.7 \text{ \AA}$. At the fully-lithiated state, initial perturbation models are the same for each NMC composition. During delithiation, Li atoms are randomly and consecutively removed from the lattice structure with a stabilized perturbation model.”

(2) Supercells (4.9 V sample) are shown in Figure 4e. What do the different colored spheres represent? How are they constructed?

Response:

The green, red, grey, blue, and purple spheres in Figure 4e represent the Li, O, Ni, Co, and Mn atoms, respectively. The corresponding annotation is added in the caption of Figure 4e in our revision.

Regarding the constructing our supercells model, for the NMC at the fully charged state of 4.9 V, about 60% Li atoms were removed compared to the fully-lithiated state. To generate the high delithiated perturbation model (~60% delithiation), we built a series of charge states with different delithiation degree from $x=0$ to $x=0.6$ in $\text{Li}_{1-x}(\text{NMC})\text{O}_2$. Let's start with the fully-lithiated state $x=0$, where the initial perturbation model please see the reply to Reviewer 2# Comment 1. (1). This preliminary perturbation model is then conducted at $T=300 \text{ K}$ with a long relaxation time to get the stabilized model. Now we randomly removed Li atoms from this stabilized model to generate $x=0.1$ model and relaxed at 300 K to construct the stabilized model subsequently. In hence, for each higher charge state, Li atoms were removed randomly from the previous one stabilized charge state. Ultimately, the 4.9 V perturbation model is constructed in a similar manner. The left panel in Figure 4e is the stabilized structure, which is obtained via relaxation at 300 K for 6 ps. The right panel in Figure 4e is the structure after thermal treatment.

Revision:

In the caption of Fig. 4, we have added the following details.

“The green, red, grey, blue, and purple spheres represent Li, O, Ni, Co, and Mn atoms, respectively.”

In the “Method” section, we have added the following details in the “Modeling Methods” part.

“During delithiation, Li atoms are randomly and consecutively removed from the lattice structure with a stabilized perturbation model.”

2. The authors need to explain how they chose the thermal treatment temperature? Is it possible to use a temperature lower than 500 K?

Response:

We thank the reviewer for bringing this up. Studying the battery cathode material under various temperature conditions is a very active research field and the selection of the temperature window often closely relates to the targeted application scenario.

For example, at low temperature, the research interest is focused on the strongly impeded lithium diffusion that is detrimental to the battery operation in harsh winter climate. The temperature of interest, therefore, is often set to -40 and 0 °C. For the cathode synthesis application, the temperature is often set to 700 to 1000 °C, because that is need for the calcination process.

For this study, there are two competing factors that drive our selection of the thermal treatment temperatures: 1) higher temperature can serve as a more effective perturbation of the lattice rearrangement; 2) lower temperature is more desirable from the energy efficiency and side-reaction-suppression perspectives. In our experiment, after a few trial-and-error attempts, we settled to 150 °C. We declare that further efforts are needed to truly pin down the optimal temperature. The optimization of the process is an on-going effort, but the fundamental mechanisms is the same with that reported in this paper.

Revision:

We have added the following discussion in the “Electrochemical Test” part of

“Methods” section.

“For this study, there are two competing factors that drive our selection of the thermal treatment temperatures: 1) higher temperature can serve as a more effective perturbation of the lattice rearrangement; 2) lower temperature is more desirable from the energy efficiency and side-reaction-suppression perspectives. In our experiment, after a few trial-and-error attempts, we settled to 150 °C. We declare that further efforts are needed to truly pin down the optimal temperature. The optimization of the process is an on-going effort, but the fundamental mechanisms is the same with that reported in this paper.”

3. More explanation and the detailed measurement conditions are needed in the caption of Figure 1b. It is very difficult to understand the history of the electrodes shown in Figure 1b.

Response:

First, all the cells were cycled once with 0.1 C (1 C= 180 mAh g⁻¹) rate at 2.8-4.9 V vs Li⁺/Li. Then they were subjected to 10 cycles with 1 C rate at 2.8-4.9 V to purposely induce lattice defects. After that, the cells were cycled once with 0.1 C rate at 2.8-4.3 V vs Li⁺/Li in order to quantify the prior-to-heating capacities. They are then charged to the targeted voltages (2.8 V, 3.7 V, 4.0 V, 4.3 V, 4.6 V, and 4.9 V) before being disassembled for the thermal treatment. After the thermal treatment, these electrodes were assembled back into coin cells and were cycled 2 cycles with 0.1 C rate at 2.8-4.3 V vs Li⁺/Li in order to quantify the post-heating capacities.

These clarifications have been added to the “Methods” section and is referenced in the caption of Figure 1b.

Revision:

In the “Electrochemical Test” part of “Methods” section and caption of Figure 1b, we add the follow details.

“First, all the cells were cycled once with 0.1 C (1 C= 180 mAh g⁻¹) rate at 2.8-4.9 V vs Li⁺/Li. Then they were subjected to 10 cycles with 1 C rate at 2.8-4.9 V to

purposely induce lattice defects. After that, the cells were cycled once with 0.1 C rate at 2.8-4.3 V vs Li⁺/Li in order to quantify the prior-to-heating capacities. They are then charged to the targeted voltages (2.8 V, 3.7 V, 4.0 V, 4.3 V, 4.6 V, and 4.9 V) before being disassembled for the thermal treatment. After the thermal treatment, these electrodes were assembled back into coin cells and were cycled 2 cycles with 0.1 C rate at 2.8-4.3 V vs Li⁺/Li in order to quantify the post-heating capacities.”

4. In page 9, line 176, I am wondering why the investigation at the single particle level equals to “at the mesoscale”. This does not seem straightforward to me.

Response:

To our knowledge, in battery research, the terminology “mesoscale” is intended for describing an intermediate length scale that bridges the nanoscale building blocks and the micro-morphology of the composite electrode. There isn’t an explicit definition with an absolute scale unit.

In the studied composite electrode, the NMC622 particles are micron-sized single-crystals. While their collective effect determines the cell-level performance, there is significant chemical complexities within each individual particles. We investigate these micron-sized particles with nano-resolution probes to reveal the structural and chemical complexity that connects the very localized electrochemical events to the large-scale electrochemical performance. Therefore, we feel that the use of “mesoscale” is appropriate in this context.

5. In the 4.9 V sample (prior-to-heating), did the authors observe layer to spinel/rocksalt transitions?

Response:

We acknowledge that an undesired phase transition from layered structure to spinel/rocksalt could occur when the cell is charged to high voltage. This has been reported in the literature [Guiliang Xu et al., Nat. Energy 4, 484-494 (2019); Feng Lin et al., Nat. Commun. 5, 3529 (2014)].

As we have described above, the severe lattice rearrangement (layered to

spinel/rocksalt transition) is not our focus because it could not be healed by the proposed mild thermal treatment.

6. In Figure 4a to 4c, the average $\Delta d/d$ actually increases after heating. Why would this happen?

Response:

We thank the reviewer for pointing it out. The rocking-curve nano-diffraction analysis provides quantitative information in relative, while the absolute values could be shifted by an arbitrary number. To verify the relative d-spacing relationship before and after heating, we summed the diffraction pattern at the rocking angle that the most number of pixels reach maximum diffraction intensities at each condition. **Figure R2** shows the two summed diffraction patterns overlay on each other at the same vicinity on the corresponding Debye-Scherrer ring, which indicates that the averaged (104) d-spacing was kept the same before and after heating. In our revision, the figure and plot for the post-heating condition in Fig. 4a and c were shifted accordingly to reflect the fact that the overall d-spacing was not changed before and after heating.

Figure R2. The summed diffraction patterns at the Bragg condition before and after heating overlay with each other on the Dybye-Scherrer ring, indicating the overall d-spacing value was kept the same during the annealing process.

7. It is recommended that the author specify the wavelength of synchrotron X-ray diffraction measurements.

Response:

The Synchrotron XRD was conducted at beamline 11-3 of Stanford Synchrotron Radiation Lightsource (SSRL) with the beam energy of 12.7 keV and wavelength of 0.975 Å. The scanning X-ray probe measurements were implemented at the hard X-ray nanoprobe beamline 3-ID of National Synchrotron Light Source II with the beam energy of 9 keV and wavelength of 1.378 Å. These information have been added to the “Method” section.

Reviewer #3:

In this manuscript, the authors studied the thermal effect on the single-crystal NMC cathode particles. The thermal healing behavior at low SOC seems quite interesting, which suggests a new strategy to extend battery cycle life. This work was done by a group of scientists with a lot of experience on battery materials and characterization techniques. The manuscript is well-written. It may be published after the authors address the following minor issues.

Response:

We are thankful for the positive assessment of our work. We also appreciate the specific and constructive comments listed below.

1. The authors need to clarify what kinds lattice defects they are trying to “heal”? Do they just mean lattice distortion and plane bending? or line/screw dislocations are also included.

Response:

The lattice defects that we are targeting are lattice distortion and plane bending, i.e., d-spacing inhomogeneity, Y-twisting, and Z-bending, which can be probed using the scanning X-ray nano-diffraction methodology. From the diffraction contrast image and the analyzed d-spacing, y and z bending maps, it is possible to distinguish certain defect types. For instance, in our previous work regarding single crystalline LiCoO₂ [Yanshuai Hong et al., Chem 6, 1-11 (2018)], the worm-shaped curves were observed (**Figure R3**), which are very similar to the edge-dislocation lines visualized in a Li-rich layered oxide particle using the Bragg coherent diffraction imaging (BCDI). In the y or z bending maps, it is possible to observe a spiral-shape vortices, which could be signature of screw dislocations. However, such evidences were not observed in the particles measured in this work, thus, we cannot specify the defect types from the available datasets and we do not intent to make such a claim.

Figure R3. Diffraction contrast images integrated over all rocking angles for a bare LiCoO_2 (LCO) and a Ti/Mg/Al co-doped LCO (TLCO) crystal. A, Diffraction contrast images integrated over all rocking angles for a bare LCO crystal. B, Diffraction contrast images integrated over all rocking angles for a TLCO crystal. The red arrows point to the worm-shape curves, which are attributed to the edge dislocation lines. The green lines highlight a different type of lattice defects, which can be attributed to the crystal twin boundaries. Figure from Yanshuai Hong et al., Chem 6, 1-11 (2018).

Revision:

In the 2nd paragraph of the “Evolution of lattice strain and Li diffusion kinetics” part, we have added the following discussions.

“It should be noted that the lattice defects that we are targeting in this study are limited to lattice distortion and plane bending, i.e., d-spacing inhomogeneity, Y-twisting, and Z-bending.”

2. Is there a specific reason to choose 150 degree C for the thermal treatment?

Response:

We thank the reviewer for bringing this up. Studying the battery cathode material under various temperature conditions is a very active research field and the selection of the temperature window often closely related to the targeted application scenario.

For example, at low temperature, the research interest is focused on the strongly impeded lithium diffusion that is detrimental to the battery operation in harsh winter climate. The temperature of interest, therefore, is often set to -40 and 0 °C. For the

cathode synthesis application, the temperature is often set to 700 to 1000 °C, because that is needed for the calcination process.

For this study, there are two competing factors that drive our selection of the thermal treatment temperatures: 1) higher temperature can serve as a more effective perturbation of the lattice rearrangement; 2) lower temperature is more desirable from the energy efficiency and side-reaction-suppression perspectives. In our experiment, after a few trial-and-error attempts, we settled to 150 °C. We declare that further efforts are needed to truly pin down the optimal temperature. The optimization of the process is an on-going effort, but the fundamental mechanisms is the same with that reported in this paper.

Revision:

We have added the following discussion in the “Electrochemical Test” part of “Methods” section.

“For this study, there are two competing factors that drive our selection of the thermal treatment temperatures: 1) higher temperature can serve as a more effective perturbation of the lattice rearrangement; 2) lower temperature is more desirable from the energy efficiency and side-reaction-suppression perspectives. In our experiment, after a few trial-and-error attempts, we settled to 150 °C. We declare that further efforts are needed to truly pin down the optimal temperature. The optimization of the process is an on-going effort, but the fundamental mechanisms is the same with that reported in this paper.”

3. In Figure 3b, the authors did not observe cracks in the 4.9 V prior to heating particle. This is somewhat surprising to me. In several previous papers, such as Pengfei Yan et al, Nature Commun 2017, 8, 14101. Intragranular fracture was observed in NMC333 particles charged to 4.7 V. Can the authors explain what led to such difference?

Response:

We thank the reviewer for bringing this work to our attention. In the referenced paper,

the intragranular cracks were observed in the NMC333 particles when charged to 4.7 V [Pengfei Yan et al., Nat. Commun., 2017, 8, 14101]. It is noteworthy that the NMC333 sample in the referenced paper was subjected to 100 cycles in the voltage range of 2.7-4.7 V. The repeated lattice expansion and contraction could clearly lead to mechanical fatigue and domain cracking. In our experiment, we focus on NMC622 samples in their early cycles. Therefore, the cracks were not observed in our experiment. In general, we fully agree with the reviewer that high-voltage cycling could be a significant contributor to the mechanical degradation of the cathode particles.

4. The authors mentioned “the localized stresses and dislocations that are closely correlated with the microcrack propagation are pervasive in single-crystalline NMC”. Did they observe dislocations and their evolution like those shown in Andrew Ulvestad et al, Science, 2015, 348, 6241?

Response:

We again clarify that the lattice defects we are targeting are lattice distortion and plane bending, i.e., d-spacing inhomogeneity, Y-twisting, and Z-bending, which can be probed using the scanning X-ray nano-diffraction methodology. From the diffraction contrast image and the analyzed d-spacing, y and z bending maps, it is also possible to distinguish certain defect types. However, the evidences for the line/screw dislocations were not observed in the particles measured in this work, thus, we cannot specify the defect types from the available datasets and we do not intent to make such a claim.

We acknowledge that coherent diffractive imaging methods [Andrew Ulvestad et al., Science 348, 1344-1347 (2015)] could offer the sensitivity to specific lattice defects, e.g. dislocations. There are pros and cons in both approaches. We have added some discussions along this line in the revised manuscript.

Revision:

In the 2nd paragraph of the “Evolution of lattice strain and Li diffusion kinetics” part,

we have added the following discussions.

“It should be noted that the lattice defects that we are targeting in this study are limited to lattice distortion and plane bending, i.e., d-spacing inhomogeneity, Y-twisting, and Z-bending. It is useful to note that coherence based imaging methods [Andrew Ulvestad et al., Science, 348, 1344-1347 (2015); Andrej Singer et al., Nat. Energy 3, 641-647 (2018)] could offer sensitivities specific to certain types of lattice defects. There are pros and cons in both approaches and a more comprehensive comparison of these methodologies is beyond the scope of this paper.”

Reviewer #4:

The authors report on a mild-annealing process at temperatures around 500 K for healing lattice strains and defects in single-crystalline NCM cathode particles. It is claimed that this method works only in specific state-of-charge regimes at voltages around 2.8 V. XAS and XANES measurements indicate that the thermal annealing at 2.8 V. does not lead to changes in the valence state of surface Ni cations, whereas at voltages around 4.9 V, annealing leads to a reduction of Ni cations in subsurface regions, which lowers the particle capacity. By means of DFT and CI-NEB calculations, activation barriers for Li transport are calculated, and it is shown that lattice healing during mild annealing enhances the Li diffusion coefficients by about two orders of magnitude. Finally, the authors suggest that during battery recycling, the mild-annealing process might be carried out as a final step after the normal high-temperature heating of NMC.

While the work is potentially very interesting for the battery community, I have serious problem in understanding the cycling protocols before and after mild annealing as well as the reported results for the capacities.

Response:

We are thankful for the overall positive assessment of our work. We also appreciate the specific and constructive comments listed below.

1. Before annealing, all cathodes were charged up to 4.9 V for several cycles. This is an extremely high potential, which should lead to massive electrolyte decomposition. In commercial batteries, NMC cathodes are never charged to such high potentials. Please comment on this.

Response:

We thank the reviewer for pointing this out. We agree with the reviewer that 4.9 V are extremely high potential, which could results in electrolyte decomposition. In addition, such a high-voltage electrochemical abuse process could also result in other detrimental side reactions, such as surface phase transition, accompanied with oxygen release, cation mixing, and active elements loss.

In our cycling protocol, we intentionally choose a high cut-off voltage of 4.9 V to promote the formation of lattice mismatch, deformation, and strain, which will persist even after the cathode is discharged to 2.8 V. Specifically, all the single-crystalline NMC622 cathodes were first charged to 4.9 V to create stresses and then cycled to different SOC's via cut-off voltage control. After the annealing process, the lattice strain in the discharged particle (2.8 V) was partially relieved as evidenced by results from scanning X-ray nano-diffraction, i.e., the disappearance of some strain edges in Fig. 4a.

Another approach to induce the lattice defect is to conduct prolonged electrochemical cycling of the cell with a mild cut-off voltage. This is more relevant to real-world commercial battery operation, however, is much more time-consuming. For the presented research, we settled to the described protocol with consideration of experimental efficiency.

2. (1) After annealing, the cathodes were charged only up to 4.3 V, and Coulomb efficiencies significantly larger than 100% were observed. The extreme case is Fig. S3e with a charge capacity close to zero for 1st post-to-heating, but a discharge capacity of 90 mAh/g.

Response:

In the initial cycle after the annealing process, the cathode was charged to 4.3 V, which is commonly adopted potential versus Li/Li⁺ to prevent the side reaction with the electrolyte. In order to study the specific capacity difference before and after annealing objectively, the cathode was charged to 4.3 V first and with an extra cycle to double check its specific capacity after annealing.

We clarify that, for investigating the annealing impact on the cathode with different SOC, the cells were disassembled at different voltage. For example, in Fig. S3e case, the cathode was charged to 4.6 V and then subjected to the annealing process. As a result, in the 1st post-heating cycle, the charge capacity is closed to zero because the as assembled cell is already in high SOC. This is indeed the reason for carrying out the 2nd post-heating cycle, which can provide a better quantification of the coulombic

efficiency after annealing. In addition, all the post-heating capacities shown in this work are based on the voltage profile of different electrodes at the 2nd cycles.

(2) Furthermore it is strange that all prior-to-heating curves from 3.7 to 4.9 show a discharge capacity around 140 mAh/g, while the curve at 2.8 V shows only 125 mAh/g. Overall, it seems problematic to deconvolute the influence of mild annealing from the influence of the distinct cut-off potentials. Please comment on these issues.

Response:

To ensure the reproducibility of our experimental results, we actually conducted several parallel experiments using the same protocol. The summary of all the measurements have been listed in Supplementary Table S2. For the case of 2.8 V electrode, 9 parallel cells were studied and all of them show a positive annealing effect. In Figure S3a, the data from cell 8 was plotted. We acknowledge the cell-to-cell variation, but the overall increment in the post-heating capacity is unambiguous.

3. The discharge capacities in Fig. S3a-f differ from those in Tab. S2, in particular for the cathode “heated at 2.8 V”. Why is this?

Response:

We are sorry for any confusion. To ensure the reproducibility of our experimental results, we actually conducted several parallel experiments using the same protocol (results shown in Supplementary Table S2).

In Figure S3, we only present arbitrary selected voltage profiles from the groups of 2.8, 3.7, 4.0, 4.3, 4.6 and 4.9 V electrodes. Specifically, Cell-8 (2.8 V), Cell-2 (3.7 V), Cell-2 (4.0 V), Cell-2 (4.3 V), Cell-4 (4.6 V), and Cell-4 (4.9 V) were plotted. All the post-heating capacities shown in Supplementary Table S2 are based on the voltage profile of different electrodes at the 2nd cycles. The corresponding explanation has been added to the caption of Supplementary Figure 3 and Table S2.

4. Furthermore, it is unclear why the material loading was only 1.5-2 mg. This

corresponds to a cathode thickness in the range of only 5 μm , much lower than the typical cathode thicknesses of 80-100 μm in typical lab-scale and commercial batteries. Please comment on this.

Response:

In this study, we focus on the defect behaviors of NMC under mild thermal stimulation from the material perspective. We agree with the reviewer that thick electrode is more relevant to the commercial cells. However, it has been reported that the thick electrode could lead to a cell polarization effect, complicating the further analysis and interpretation. Therefore, we intentionally fabricate thin electrode to simplify the study.

5. Minor points: (1) What is the origin of Ni cation reduction at high voltages of 4.9 V? Release of molecular oxygen? (2) Page 9: The authors use the term “operando TXM”. However in this case, the method was not used in operando.

Response:

(1) We agree with the reviewer that, the origin of Ni cation reduction at high voltage of 4.9 V can be ascribed to the molecular oxygen release, which is confirmed by the EXAFS fitting results for the structural parameters around Ni atoms (Supplementary Figures 5-7). Specifically, as shown in Fig. 2d and Supplementary Table S4, the Ni-O coordination number was decreased to 5.0 for the 4.9 V electrode (before thermal treatment) compared to that of 6.0 for the 2.8 V electrode.

(2) The “operando” here means operating TXM measurement continuously during the thermal control. However, to avoid any confusion, we have deleted “operando” in our revision.

6. In summary, major revisions of the manuscript are needed, before a decision about the acceptance can be made.

Response:

We are thankful for the thorough assessment of our work. We look forward to further interaction with the reviewer 4#.

General response and remarks:

We would like to take this opportunity to thank all the unnamed referees again for the invaluable advice and suggestions. We have tried our best to address all the questions from the reviewers. We believe that this review process has significantly improved the quality of our paper and we look forward to hearing any further comments from the reviewers and the editorial office.

Yijin Liu, Kejie Zhao, Linsen Li and Jieshan Qiu

REVIEWERS' COMMENTS

Reviewer #2 (Remarks to the Author):

The manuscript can be accepted in present form.

Reviewer #3 (Remarks to the Author):

The authors have addressed all my concerns. This version can be accepted for publication.

Point-to-point revision summary for

Article NCOMMS-21-35847A–*Thermal-healing of lattice defects for high-energy single-crystalline battery cathodes*

Reviewer #2:

The manuscript can be accepted in present form.

Response:

We are thankful to the reviewer for their efforts in evaluating our submission and for making constructive comments, which we find very useful for our revision. We are delighted that the reviewer has recommended the acceptance of our manuscript for publication in Nature Communications.

Reviewer #3:

The authors have addressed all my concerns. This version can be accepted for publication.

Response:

We are thankful to the reviewer for the invaluable advice and suggestions. We are delighted that the reviewer has recommended the acceptance of our manuscript.